# A Dynamic Entropy Approach Reveals Reduced Functional Network Connectivity Trajectory Complexity in Schizophrenia

**DOI:** 10.3390/e26070545

**Published:** 2024-06-26

**Authors:** David Sutherland Blair, Robyn L. Miller, Vince D. Calhoun

**Affiliations:** Tri-Institutional Center for Translational Research in Neuroimaging and Data Science (TReNDS), Georgia State, Georgia Tech, Emory University, Atlanta, GA 30303, USAvcalhoun@gsu.edu (V.D.C.)

**Keywords:** functional network connectivity, dynamic functional connectivity, Shannon entropy, NeuroMark, independent component analysis, principal component analysis, sliding window correlation, multiple linear regression

## Abstract

Over the past decade and a half, dynamic functional imaging has revealed low-dimensional brain connectivity measures, identified potential common human spatial connectivity states, tracked the transition patterns of these states, and demonstrated meaningful transition alterations in disorders and over the course of development. Recently, researchers have begun to analyze these data from the perspective of dynamic systems and information theory in the hopes of understanding how these dynamics support less easily quantified processes, such as information processing, cortical hierarchy, and consciousness. Little attention has been paid to the effects of psychiatric disease on these measures, however. We begin to rectify this by examining the complexity of subject trajectories in state space through the lens of information theory. Specifically, we identify a basis for the dynamic functional connectivity state space and track subject trajectories through this space over the course of the scan. The dynamic complexity of these trajectories is assessed along each dimension of the proposed basis space. Using these estimates, we demonstrate that schizophrenia patients display substantially simpler trajectories than demographically matched healthy controls and that this drop in complexity concentrates along specific dimensions. We also demonstrate that entropy generation in at least one of these dimensions is linked to cognitive performance. Overall, the results suggest great value in applying dynamic systems theory to problems of neuroimaging and reveal a substantial drop in the complexity of schizophrenia patients’ brain function.

## 1. Introduction

Schizophrenia (literally: *split mind*) ranks among the most studied disorders in modern psychiatry and neuroscience. There are manifold reasons for this, but perhaps the most prominent is the sheer variety of symptoms which it can produce. The symptoms of schizophrenia broadly fall into three categories, positive, negative, and cognitive, and can range from delusion, paranoia, and hallucination to apathy, anhedonia, and social withdrawal. This makes schizophrenia a difficult disorder to treat, as treatments for one group of symptoms may have limited or no effect on another. Indeed, it has been suggested that schizophrenia should not be considered a single disorder at all but rather a family of disorders aggregated by historical precedent [1].

For the past quarter of a century, hypotheses on schizophrenia’s causes have focused on the interactions between brain regions. Perhaps the most prominent is the dysconnectivity hypothesis [2], which may be summarized as the proposition that aberrant connectivity, rather than focal abnormalities, is the primary cause of schizophrenia. In this sense, it may be considered a forerunner to the concept of distributed cognition—quite an early forerunner, as it was first proposed at the beginning of the 20th century [3,4,5]. The past half a century of research into schizophrenia has amassed considerable evidence in its favor [6], with experimental results suggesting that schizophrenia affects connectivity and plasticity from the level of individual synapses [7,8,9,10,11] to interregional communication [12,13,14] and white matter tracts. Most of these studies, however, base their analyses on anatomically rather than functionally derived atlases and do not examine time courses through a latent or state space. They may, therefore, miss some of the crucial insights which examining such systems through these lenses can provide.

At the “global”, or whole-brain, scale, most studies have examined static functional connectivity (FC) or structural connectivity (SC). Such studies have reported reduced global connectivity in schizophrenia patients [15,16], particularly between auditory, sensorimotor, and visual networks [17]. In addition, network-based analyses of functional connectivity in schizophrenia indicate a general reduction in organization and efficiency in the structural and functional connectivity of schizophrenia patients [18]. Beyond this, however, findings have been inconsistent. Some reports suggest decreased communication between the frontal and temporal areas of the brain [12]; others have found increased connectivity within the default-mode network [19]; still, others imply decreased connectivity within and between the default-mode network and cortical regions [20]. Attempts to model the static functional connectivity of schizophrenia patients have provided equally confused results, with both reduced effective connectivity and increased structural connectivity suggested [21]. Overall, then, the analysis of structural and static functional connectivity has been unable to conclusively identify the large-scale changes which underlie the symptoms of schizophrenia.

Statical analyses’ neglect of brain dynamics may be one reason for these inconclusive results. Electroencephalography (EEG) has demonstrated that functional microstates constantly and fluidly change [22,23,24], a fact captured in fMRI in 2010 [25,26]. Although fMRI captures microstate alterations at a timescale of seconds rather than the milliseconds of EEG, its superior spatial resolution compared to EEG has allowed researchers to precisely identify recurring connectivity states [15,27,28,29,30]. Schizophrenia patients have displayed altered dynamics in these states. For instance, a 2014 study suggested that patients have a higher probability of entering states with attenuated cortical–subcortical connections and increased intrasensory connectivity than controls, and that these same patients also display elevated low-frequency power in thalamosensory communication [17]. This same study suggested that the thalamosensory hyperconnectivity reported in its static FC analysis may result from these alterations. However, aside from state transition probabilities and average dwell times, this study did not report any metrics designed to capture state dynamics. This lack of metrics designed to capture the dynamics of functional connectivity has proven to be a problem, not just for schizophrenia research but also for the broader field of functional neuroimaging.

We seek to add a means of capturing functional connectivity dynamics to the neuroscience toolkit, specifically a means of measuring the entropy rate of individual fMRI scans [31]. In the present article, we use this framework to demonstrate a statistically meaningful difference in functional connectivity entropy in a dataset of medicated schizophrenia patients and demographically matched controls. Rather than attempting to explicitly track interactions between brain regions, this framework defines a basis for a state space within which to plot each subject’s time course. This approach allows for the direct application of tools and metrics from dynamic systems analysis and information theory to subjects’ time courses and thus reduces the challenge of comparing subject trajectories. The decision to quantify fMRI dynamics in a state space is not novel [32], but the present framework specifically seeks to make efficient use of the Shannon entropy. The results of this analysis suggest substantially disordered state dynamics in the patient population, with patients displaying simpler trajectories in five out of eight state space dimensions, one of which correlates with cognitive performance. Previous studies in EEG have found similar entropy deficits in patients [33], but EEG’s poor spatial resolution has precluded any detailed mapping of the connectivity state space. Overall, this represents a positive step towards quantifying the dynamic connectivity alterations which underlie this psychiatric disease and towards the discovery of a clinically useful basis space in which to diagnose and predict patient treatment.

## 2. Materials and Methods

A high-level overview of the pipeline is displayed in Figure 1. Detailed explanations of each step follow.

### 2.1. Data Collection

This study utilizes control and schizophrenia patient data from the Function Biomedical Informatics Research Network (FBIRN) repository [34], preprocessed according to the description given in [35]. To summarize, a statistical parametric mapping package (SPM12) was used to correct for subject head motion and slice timing differences, to warp subject anatomy to the Montreal Neurological Institute (MNI) echo planar imaging (EPI) template space, to resample the collected data to 3×3×3 mm^3^ isotropic voxels, and to smooth the resampled fMRI images with a Gaussian kernel with a full width at half maximum (FWHM) of 6 mm. Subjects with head motion greater than 3° were excluded from the study, as were subjects whose full brains could not be normalized due to incomplete imaging data. These criteria led to a final dataset of 151 schizophrenia (SZ) patients and 160 healthy controls (HCs).

### 2.2. Estimation of the Spatial Functional Networks

Spatial functional networks were estimated using NeuroMark’s adaptive independent component analysis (adaptive ICA) [35], which extends spatially constrained independent component analysis [36,37] to map known fMRI network templates to novel subject data. This requires balancing two competing goals at a time: to maximize the spatial independence of networks in each subject and to ensure that the network maps in each subject correspond to known group-level templates. Here, we use the multi-objective optimized ICA with the reference (MOO-ICAR) approach, which maximizes two competing objective functions in turn until a solution is achieved. This allows adaptive ICA to capture subject-unique characteristics while maintaining comparable functional networks across datasets. It should be noted that this method allows us to capture both the internal structure of brain functional connectivity networks and the extent of inter-network connectivity via static and sliding-window functional connectivity estimates.

### 2.3. Estimation of the Functional Network Connectivity

Before estimating the functional network connectivity (FNC), Du et al. [35] chose to remove noise sources from each functional network’s subject-level time series. The removal of noise sources involved four steps: first, the removal of linear, quadratic, and cubic trends in the data; second, multiple regressions of the six realignment parameters and their temporal derivatives to control for in-scanner motion; third, de-spiking to remove outliers; and fourth, band-pass filtration to select for signals in the 0.01–0.15 Hz frequency bands. Once these steps were completed, subject-level static functional network connectivity (sFNC) was computed via Pearson correlation. Other measures of statistical similarity could have been used; for instance, mutual information has been proposed due to its sensitivity to nonlinear interactions [38,39]. However, Pearson correlation’s simplicity, interpretability, and ease of computation means it remains the dominant method for estimating functional connectivity.

While the static FNC provides valuable information on the extent of inter-network communication, its poor time resolution makes it unable to capture the dynamics of this communication. The two most notable methods proposed to circumvent this problem are the sliding time window approach [25,27,30] and coherence-based connectivity [40,41,42,43]. The present study uses the sliding window approach. As the name suggests, this method slides a window over the time series of each ICN in small steps, thus segmenting the total time series into many short, overlapping time series. The functional network connectivity of each time series window is computed in the same way as for static FNC, and the resulting N×N connectivity matrices are concatenated into an N×N×T array (N being the number of functional networks and T the number of time series windows). This study convolved a normal distribution with a mean of zero and a standard deviation of three $\left(N\left(\mu =0,\;\sigma =3\right)\right)$ with a rectangle 40-times-to-repetition (TRs) long [35] to generate its selection window.

### 2.4. Number of Temporally Independent Sources

Most clustering or source separation algorithms require users to specify the number of sources for which the algorithm should search. This poses a problem in neuroimaging analysis, as the number of recurrent connectivity states which the human brain expresses has not been fully established. As such, researchers do not know a priori the appropriate model order to input into their separation algorithm. It is common to circumvent this problem by testing several numbers of recurrent states, usually within the range of four to eight [44,45,46,47,48], and determining the optimal number by applying a validity index or comparing the results of runs with different source counts. In this instance, an eight-source model was found to maximize the number and size of group-level differences while maintaining a reasonable model order.

### 2.5. Temporal Functional Network Connectivity Profiles (tFNCPs)

The goal of this study is to identify recurring, overlapping, temporally invariant functional brain patterns and to capture their dynamics. Both clustering and source separation algorithms may be used to achieve this goal, dependent on how these states are hypothesized to manifest in time-resolved fMRI images. A clustering algorithm, such as *k*-means, may be appropriate if each time-resolved fMRI “snapshot” primarily consists of a single state, S. If, on the other hand, each time-resolved snapshot contains a mixture of states, S, a blind source separation algorithm may be more applicable. As this study presumes that each time-resolved fMRI image consists of a linear mixture of underlying “source” connectivity states, S, a linear source separation algorithm is required. In addition, an entropy-based analysis is made considerably simpler when the time courses, TC, are statistically unrelated. As such, we required a linear source separation algorithm which maximizes the statistical independence of each predicted source. Independent component analysis (ICA) is the obvious solution to this problem, as it has been proven to minimize the statistical dependencies between sources in neuroimaging data [49] and to isolate functionally meaningful communities in spike train data [50].

Most ICA algorithms accept two-dimensional inputs, with one dimension representing input variables and the second representing samples. For instance, the time series of N functional networks should produce a data array of N×T, with T being the number of samples taken over the course of recording. A dFNC array, on the other hand, has three dimensions, N×N×T, which standard ICA algorithms cannot process. We circumvent this by converting the upper triangle of each sample dFNC$\left(t\right)$ array to a vector. Repeating this process across all samples converts the data array from N×N×T to $\frac{N\left(N-1\right)}{2}\times T$, which can be decomposed into maximally independent sources by any standard ICA algorithm.

Before maximizing independence, researchers typically whiten the data via principal component analysis (PCA). This serves to reduce the dimensionality of the input data and to improve the estimation of independent components, both of which optimize and stabilize ICA outputs. We ran the InfoMax ICA algorithm [51,52,53] 150 times and consolidated the results via the ICASSO method [54] to identify repeatable tFNCPs and time courses.

### 2.6. Entropy Analysis

We used a Kozachenko–Leonenko entropy estimator to estimate each state’s subject-level rate of entropy production:$$H_{k,N}=m\ln{\bar{\rho }}_{k}+\ln\left(N-1\right)-\psi \left(k\right)+\ln c_{1}\left(m\right)$$
where ${\bar{\rho }}_{k}$ is the geometric mean of the distance between the kth nearest neighbors in the signal:$$\rho \left(x,y\right)=\sqrt{\sum_{j=0}^{m}{\left(x_{j}-y_{j}\right)}^{2}}$$
$$\rho_{i,k}≔min\left\{\rho \left(X_{i},X_{j}\right)\right\},\;j\in \left\{1,\ldots ,N\right\}\backslash \left\{i,\;j_{1},\ldots ,j_{k}-1\right\}=\rho \left(X_{i},X_{j_{k}}\right)$$
$${\bar{\rho }}_{k}=\sqrt[{N}]{\prod_{i=1}^{N}\rho_{i,k}}$$
and $\psi \left(k\right)$ is the digamma function:$$\psi \left(k\right)=\frac{\Gamma^{\prime }\left(k\right)}{\Gamma \left(k\right)}=\frac{d}{dk}\ln\Gamma \left(k\right)=\int_{0}^{\infty }\left(\frac{e^{-t}}{t}-\frac{e^{-kt}}{1-e^{-t}}\right)dt,$$
according to refs. [55,56,57,58]. This produces an $S\times N_{subjects}$ array of entropy rate values, with S being the number of sources and $N_{subjects}$ the number of subjects. The substates’ temporal independence allows us to estimate the joint entropy of all sources by summing the entropy rate across substates [59]:$$H\left(C_{1},\ldots ,C_{N}\right)=\sum_{j=1}^{N}H\left(C_{j}\right)$$Note that this calculation requires that source signals be statistically independent. Dependencies between sources must be accounted for when estimating source entropy, which makes evaluating joint entropy quite difficult when dealing with statistically related signals. ICA’s minimization of intersource relationships was a major motivation for its selection.

### 2.7. Comparison Tests

We utilized both a difference-of-means permutation test [60] and the Kolmogorov–Smirnov two-sample test to search for differences between groups. Student’s *t*-test was also employed when the Jarque–Bara test indicated normally distributed data. Multiple-comparison correction consists of the false discovery rate [61].

### 2.8. Regression Analysis

To correct for possible confounds in the data, a multiple linear regression was employed alongside standard hypothesis tests. This regression separately modeled the effects of site, age, gender, and diagnosis on subject-level joint and source entropies. Additional regression analyses examined the effects of clinical (PANSS positive and negative) and cognitive (CMINDS composite) scores on subject entropy distributions, again while correcting for site, age, and gender effects. As cognitive scores are highly correlated (Figure 2), it was decided to examine only the most general score, namely the CMINDS composite score [62], to avoid effect cross-contamination.

**Figure 1 entropy-26-00545-f001:**
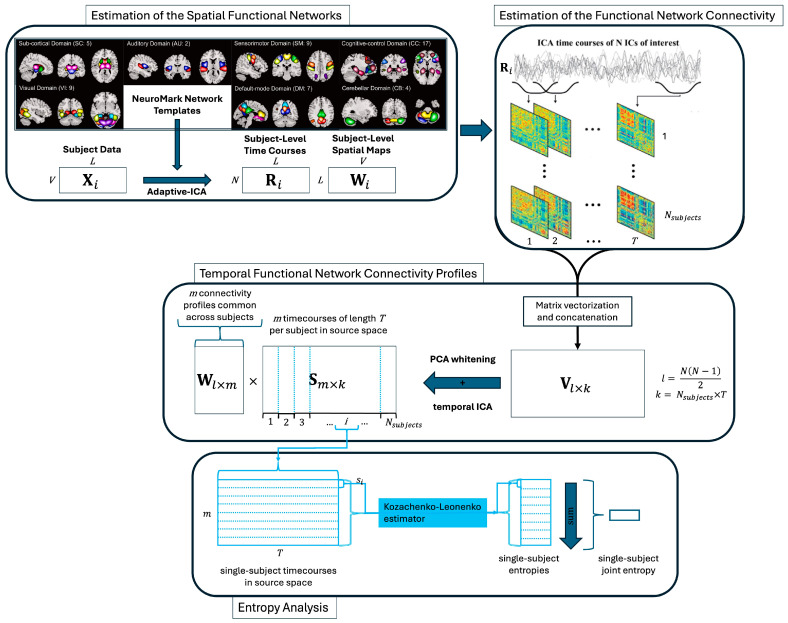
Procedure for identifying subject-level tFNCP and joint entropy. Procedural schematic for identifying subject-level tFNCP and joint entropy in functional magnetic resonance images (fMRIs). Adaptive ICA [37] is used to map V subject-level voxel time courses of length L to N preestablished network templates [35] $\left(N=53\right)$. This produces subject-level time courses and spatial maps which maintain comparability. Sliding-window correlation [25] then finds the time-resolved correlation of each subject to produce N×N×T connectivity arrays (with a window length of 40 TRs). The upper triangle of each connectivity matrix is vectorized to produce $N_{subjects}$ l×T subject-level time-resolved connectivity records, which are concatenated to produce a single l×k record of subject connectivities $\left(l=\frac{N\left(N-1\right)}{2},\;k=N_{subjects}\times T\right)$. After principal component analysis (PCA) whitens and reduces this array to m spatial dimensions, ICA is used to extract m temporally independent source time courses $\left(S\right)$, along with the coefficient vectors which map from connectivity to source space $\left(W\right)$. Each subject’s time courses are then isolated for entropy analysis.

A version of the Kozachenko–Leonenko estimator [56,57,63] was employed to estimate the entropy rate for each subject. Specifically, in each subject, this estimator computed the entropy rate of each source, $s_{i},$ over the course of the scan. The temporal independence which ICA enforces between sources ensures that the entropy rate of source i has minimal effect on the entropy rate of source j $\left(i\neq j\right)$. The joint entropy rate of each subject may thus be estimated by summing the subject-wise entropy rates of all sources.

## 3. Results

### 3.1. Static Functional Network Connectivity

As a pre-analysis sanity test, we compared static functional network connectivity (sFNC) in the same fashion as [35] for comparison purposes. Figure 3A displays the mean sFNCs of the FBIRN dataset. Notably, the healthy controls display visibly greater mean absolute FNC values than the schizophrenic patients across the majority of the sFNC matrices’ $\frac{N\left(N-1\right)}{2}$ correlation coefficients. This is due to a reduction in the sFNC variance in patients, visible in bar chart format (Figure 3C) and in the absolute difference between the control and patient means (Figure 3D,E). This visible reduction in patient sFNC values provides initial support for the dysconnectivity hypothesis, as it suggests that schizophrenia is broadly characterized by reduced internetwork communication compared to the controls. This, in addition to the block-diagonal structure of the sFNC matrices, suggests that the data have been correctly reconstructed.

### 3.2. Joint Entropy

We estimate each tFNCP’s subject-level entropy rate with the variation in the Kozachenko–Leonenko estimator [57,58] based on the *k*-nearest-neighbor distances between sample points [56,63]. This estimation algorithm is necessary due to the unavailability of source signals’ complete probability distributions, which is necessary for an exact calculation of signal entropy. Fortunately, the statistical independence of the tFNCPs prevents any dependence between their respective entropy rates. As such, the joint entropy rate of each subject can be estimated simply by summing all tFNCPs’ entropy rates within that subject. The resulting group-level distributions of the dynamic functional connectivity’s joint entropy are displayed in Figure 3B, with schizophrenia patients on the left and healthy controls on the right. Patients display depressed joint entropy rate relative to the controls, with the Kolmogorov–Smirnov test demonstrating that this elevation is highly significant $\left(p=4.506\times 10^{-8},\;D^{*}=0.2188\right)$. A difference-of-means permutation test confirms this finding $\left(p=9.999\times 10^{-5},\;G=-0.4095\right)$. Student’s t-test was not employed because the joint distribution fails the Jarque–Bara test, thus violating Student’s assumption of normality. Group-level means and standard deviations are listed in Table 1.

### 3.3. tFNCP Entropy

Having confirmed that subject joint entropy rate differs between populations, an obvious question is whether these differences are evenly distributed amongst tFNCPs or whether they are concentrated within a subset of them. To answer this question, we compared the group-level entropy rates of each tFNCP using the Kolmogorov–Smirnov two-sample test, with the false discovery rate [61] correcting for multiple comparison. A Student’s *t*-test was also applied to the entropy rate distributions, which passed the Jarque–Bera test for normality. This approach reveals decreased patient entropy rates in five of eight tFNCPs (Figure 4), with no qualitative differences between the Kolmogorov–Smirnov two-sample test and Student’s *t*-test results. Figure 4 displays the connectivity matrices and entropy distribution boxplots of each tFNCP, with group entropy means, variances, and test statistics listed in Table 2.

### 3.4. Multiple Linear Regression

The wide range of ages and the seven collection sites of the FBIRN dataset raise the possibility of substantial age and site effects, potentially large enough to affect the outcome of the previously described hypothesis tests. To account for this, we employed a multiple linear regression analysis to separate the effects of site, age, gender, and diagnostic status (“patient” vs. “control”) on subjects’ joint and tFNCP-level entropy rate. We used similar analyses to examine the link between subject-level entropy, clinical scores, and cognitive scores. Clinical scores consist of the PANSS positive and negative scores, while the CMINDS composite score is used as a surrogate for cognitive scores due to the high correlation among the subscales (Figure 2).

#### 3.4.1. Diagnostic Effects

The multiple linear regression diagnostic results, after controlling for site, age, and gender, remain consistent with the statistical hypothesis tests. The site effects are both substantial and highly significant, but they do not change the outcomes of the hypothesis tests. Age and gender effects were found to be small and statistically insignificant.

#### 3.4.2. Symptomatic Effects

The multiple linear regression identified no statistically significant relationship between the patients’ PANSS scores and entropy rate in any of the tFNCPs. The effect sizes are small, with only four of eighteen examined relationships displaying effect sizes of the order $10^{-2}$ or higher.

#### 3.4.3. Cognitive Effects

One tFNCP, number five (5), displays a small but highly significant $\left(\beta =0.0633,\;p=0.0053\right)$ relationship between the subjects’ CMINDS composite score [62] and entropy rate. Interestingly, this tFNCP is characterized by strong anticorrelations between two higher cognitive domain regimes: default-mode and subcortical networks oppose visual networks and sensorimotor networks, with cognitive control networks split between these two regimes. The default mode is the most prominent of these domains, being strongly anticorrelated with both opposing domains. The joint entropy rate is also positively related to the CMINDS composite score, although this relationship does not reach statistical significance $\left(p=0.1203\right)$.

## 4. Discussion

This article presents a novel method for evaluating the entropy of the dynamics of temporally independent FNCs and the relationship of that entropy to psychiatric disease. Of the eight temporally independent FNC profiles identified, five display significant entropic reductions in schizophrenia patients, with the remaining three showing no such reduction. One of these five diagnostically variable tFNCPs displays a significant positive relationship with cognitive score, with this single tFNCP including strong contributions from default-mode networks, cognitive control networks, and other higher cognitive domains.

The initial inspection of the group-level static functional network connectivity broadly supports previous work showing reduced global connectivity in schizophrenia patients [64]. Positive correlations between auditory, sensorimotor, and visual networks appear notably attenuated in the patients compared to their healthy counterparts. This reduced connectivity strength and structure may reduce patients’ ability to rapidly reconfigure network relationships [65,66].

Extant research on dynamic functional connectivity consistently suggests an attenuation in the complexity of connectivity dynamics in psychiatric disorders, particularly in schizophrenia. Prior work by Miller et. al. [32] has shown that patients explore less of their connectivity state space than comparable healthy controls do and that their trajectories within this state space are both more rigid and more repetitive than controls as well. These findings imply a simplification of dynamics in patients, in line with the reduction in entropy described in this article. Such simplification may underlie the cognitive rigidity and inflexibility known to characterize some psychiatric disorders. Interestingly, the findings of Miller et al. [32] and the present article contrast with research by Prof. Kringelbach of Oxford University, as Kringelbach et. al. found that the intake of psychedelic substances increases activity diversity, turbulence, and information flow compared to placebos [67,68,69]. This suggests that hallucinatory experiences in the psychiatric and psychedelic realms possess fundamentally different operational mechanisms. Future research should attempt to identify the mechanisms in each of these populations. The finding of reduced entropy in patients may also align with the finding that psychiatric patients operate in a less metastable regime than their healthy counterparts [70,71]. Similarly, it may imply that patients occupy a reduced functional manifold compared to controls [72]. The concentration of entropy declines in specific tFNCPs may suggest an alteration in the connectivity’s energy landscape as well, as described in [69]. Which, if any, of these hypotheses can best predict the subject trajectories in the tFNCP basis space may prove a valuable line of research.

Of the eight tFNCPs identified in this study, five display significant, positive, linear relationships between the entropy rate and the diagnosis of schizophrenia. One of these further displays a small but highly significant positive relationship between general cognitive function (CMINDS score) and entropy rate. Within these five tFNCPs, the most notable feature of the functional domain structure is the stability of the visual–sensorimotor domain block. In four of the five tFNCPs with significant entropy–diagnosis relations, these domains strongly correlate with one another. Further, of these four tFNCPs, three extend the block to include the auditory domain. Only the eighth tFNCP does not display a strong visual–sensorimotor domain block. Instead, the eighth tFNCP mostly displays control and visual domain interactions with multiple other domains, mostly visual.

Of the five tFNCPs with altered entropy in schizophrenia patients, each displays a different block to anticorrelate, or oppose, the dominant visuo-sensorimotor–auditory domain alignment. Cerebellar and subcortical networks primarily make up the fourth tFNCP’s opposition, along with most of the default-mode and cognitive control domains. Three default-mode and cognitive control networks align with the visuo-sensorimotor blocks as well. The fifth tFNCP, which appears to influence the CMINDS score, recruits the subcortical and default-mode networks in opposition to the sensorimotor–visual–auditory block. The seventh tFNCP appears to possess three distinct domain-level modules, with cognitive control and default-mode networks correlated with themselves, while subcortical networks oppose the sensorimotor–auditory block. The eighth tFNCP, as noted, shows links between the cognitive control, default-mode, and visual domain areas.

When observing all eight tFNCPs as well as the diagnostically significant five, two notable trends emerge. First, no tFNCP displays a strong intra-domain correlation. High correlation values are instead concentrated between domains, as seen in the cerebellar–subcortical block of the third tFNCP (3). In fact, only two tFNCPs display high intradomain correlation: numbers 4 and 5. Even within these tFNCPs, this is the exception rather than the rule, with tFNCP 4 showing high intra-domain correlation only in the sensorimotor and visual domains and tFNCP 5 only in the default-mode networks. This consistent lack of strong intra-domain correlation stands in notable contrast to the static FNC of both groups, the strongest connections of which are all within domains.

The second notable trend across all eight tFNCPs is that cognitive control networks seem to straddle the line between the sensorimotor–visual–auditory block and its opposition. This domain is generally divided between more evident functional modules, but its strongest correlations consistently tend towards the sensorimotor–visual block, even in cases where most of its networks align against it. Its nonaligned status is most apparent in the seventh tFMCP, in which the cognitive control and default-mode networks form a third block which does not largely correlate with the sensorimotor or cerebellar functional modules. The cognitive control domain also displays a distinct tendency to split into two largely separate networks, with relatively weak autocorrelation and strong correlation or anticorrelation with other domains being present in four of the five significant tFNCPs.

## 5. Conclusions

Overall, this article proposes a novel means of quantifying entropy in neuroimaging signals. Its basis in well-established methods for identifying functionally relevant spatial maps [35,42,73,74] and maximizing the independence of temporal signals [49] lends credence to its findings of reduced dynamic connectivity trajectory complexity in schizophrenia patients. This reduction appears specific to identifiable tFNCPs, one of which significantly relates to cognitive function. It thus represents a potential link between the fields of neuroimaging and psychological analysis and is a substantial step in the ongoing search for biomarkers of psychiatric disease.

Future work with this and related methods should focus on bridging the gap between computational psychiatry, neuroimaging, and dynamical systems models. The ability to directly identify the time courses and complexity of specific tFNCP trajectories means that similar methods may be able to complement theoretical work on cognitive manifolds [72] or confirm extant studies on the transition networks and energy landscapes of the human chronnectome [75,76,77,78,79]. Similar methods may also be able to identify functionally related subgroups within heterogeneous disease categories, as has been achieved with certain psychological models [1,80]. This, in turn, could lead to the development of biomarkers of symptom dimensions directly, which may provide a more reliable and intuitive classification system for psychiatric distress than the extant Diagnostic and Statistical Manual of Mental Disorders (DSM-V) or International Classification of Diseases (ICD). We thus anticipate extensive use of this and related methodologies in future studies as part of the wider quest to catalog and diagnose psychiatric and neurological dysfunction.

## Figures and Tables

**Figure 2 entropy-26-00545-f002:**
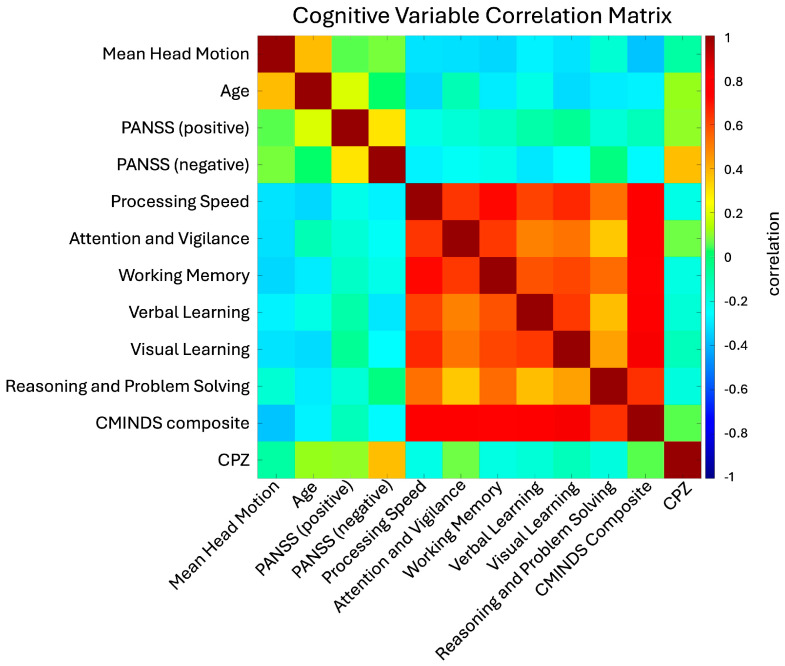
Clinical and cognitive variable relations. Correlation matrix of the clinical and cognitive variables assessed in the FBIRN dataset. The primary result is the consistently high correlation between cognitive scores (processing speed through the CMINDS composite score). This high correlation informed the decision to only include the CMINDS score in our linear regression analysis, as including other cognitive variables would likely split the effect while providing little additional information. The low correlation between negative and positive PANSS scores is also of note, as it suggests considerable heterogeneity in subject symptom expression.

**Figure 3 entropy-26-00545-f003:**
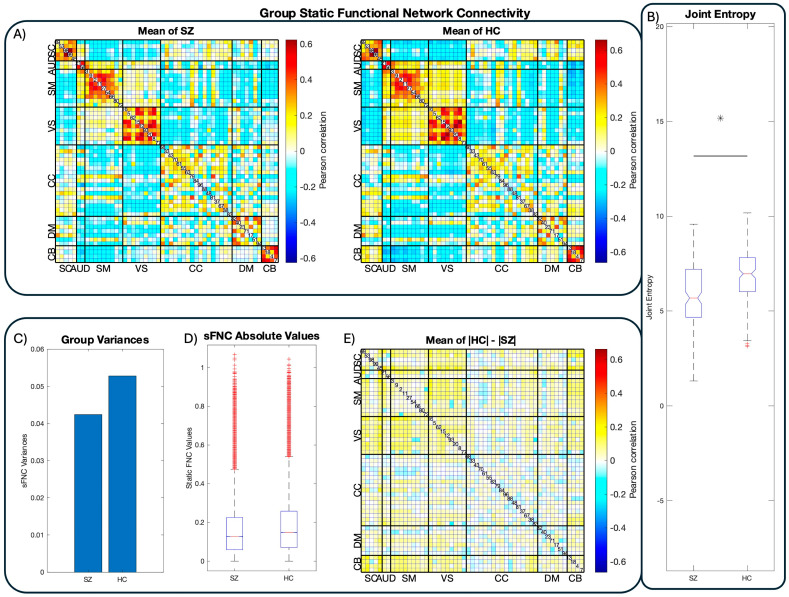
Group-level FNC and joint entropy. The results from group-level functional network connectivity (FNC) and joint entropy estimates. Panel (**A**) displays the connectivity matrices of the respective groups after mapping preprocessed fMRI images to the NeuroMark atlas [35]. Functional networks (FNs) are ordered and labeled according to their functional domains. Panel (**B**) displays the joint entropy distributions of each group’s dynamic functional network connectivity in boxplot format. In line with previous work [32], the healthy controls display substantially higher joint entropy than the schizophrenia patients, suggesting reduced flexibility in patient connectivity and computation. Visual inspection suggests that schizophrenia patients’ functional connectivity is generally dampened compared to healthy controls, in line with the dysconnectivity hypothesis and its supporting work. Panels (**C**–**E**) confirm this supposition by demonstrating increased variance in control static FNC and the fact that the controls show higher average sFNC magnitudes than patients.

**Figure 4 entropy-26-00545-f004:**
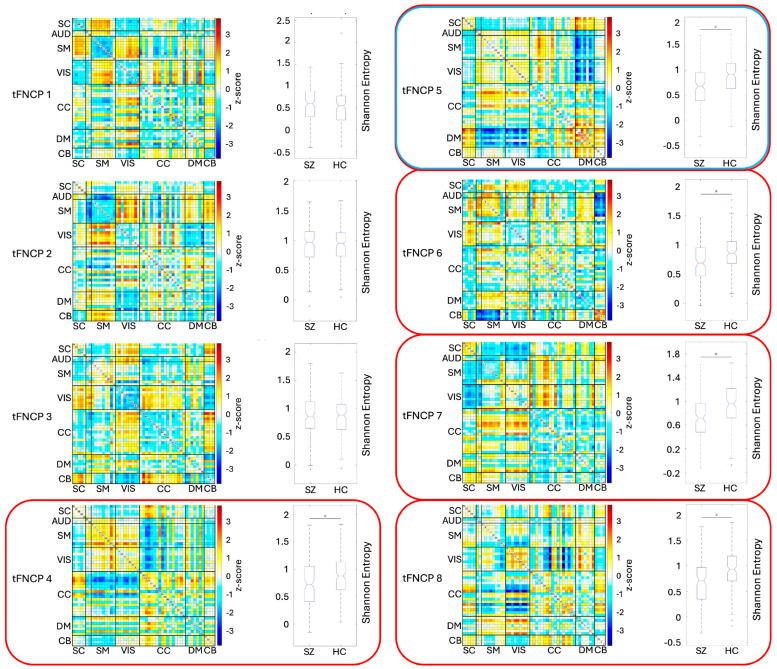
tFNCP-level maps and entropies. Our study finds eight tFNCPs shared between the controls and schizophrenia patients. Five of these eight, outlined in red above, display significant group-level entropy rate changes. The entropy rate of the fifth tFNCP, outlined in red and blue, also positively relates to the composite CMINDS cognitive score. Spatial maps of these tFNCPs are displayed in connectivity matrix format, with the respective tFNCP entropy distribution shown in boxplot format. Both regression and statistical tests show that the patients exhibit significantly attenuated entropy in five tFNCPs. It is noteworthy that of these five tFNCPs, three display stable intra- and inter-domain correlation between the visual, auditory, and sensorimotor domains. In all three cases, this block is strongly anticorrelated with at least one other domain: (part of) the cognitive control domain, the default-mode domain, or the cerebellar domain. The cognitive control domain also displays a distinct tendency to split into two largely separate networks, with weak autocorrelation and strong correlation or anticorrelation with other domains being present in four of five significant tFNCPs. Somewhat oddly, no tFNCP displays strong intra-domain correlation. High correlation values are instead concentrated between domains, as seen in the cerebellar–subcortical block of the third tFNCP (3). In fact, only two tFNCPs display high intradomain correlation: numbers 4 and 5. Even within these tFNCPs, this is the exception rather than the rule, with tFNCP 4 showing high intradomain correlation only in the sensorimotor and visual domains and tFNCP 5 only in the default-mode networks.

**Table 1 entropy-26-00545-t001:** Group-level means, variances, and differences $\left(\beta \right)$ of tFNCP and joint entropies. Note that the difference between patient and control entropies reverses after the third tFNCP (3), with control entropies becoming consistently greater than patient entropies. Note also that the magnitude of β increases substantially in the last five tFNCPs. Since controls were assigned a contrast value of 0 and patients a contrast value of 1, a negative β indicates depressed entropy in patients.

	Mean ± Variance (Patients)	Mean ± Variance (Controls)	*β* ± Standard Error
tFNCP 1	0.60064 ± 0.17609	0.53301 ± 0.15494	0.04467 ± 0.04456
tFNCP 2	0.93204 ± 0.10030	0.93372 ± 0.09301	0.00388 ± 0.03498
tFNCP 3	0.88852 ± 0.13293	0.84955 ± 0.10751	0.03043 ± 0.03877
tFNCP 4	0.71817 ± 0.18799	0.89108 ± 0.12927	−0.16376 ± 0.04556
tFNCP 5	0.65917 ± 0.16392	0.88848 ± 0.12678	−0.23097 ± 0.04298
tFNCP 6	0.69955 ± 0.11751	0.86619 ± 0.08971	−0.17171 ± 0.03671
tFNCP 7	0.72430 ± 0.13709	0.93682 ± 0.13044	−0.20885 ± 0.04164
tFNCP 8	0.67983 ± 0.18125	0.94130 ± 0.13691	−0.25684 ± 0.04519
Joint Entropy	5.90221 ± 2.58525	6.84015 ± 1.98335	−0.95316 ± 0.17359

**Table 2 entropy-26-00545-t002:** Summary of linear regression and hypothesis test results from examining the relation of entropy to subjects’ diagnostic status. Controls were assigned a group contrast label of 0, while patients were assigned a group contrast label of 1. All applied tests produced similar results, namely highly significant relations between diagnostic status and the entropy rate of the last five tFNCPs. The directionality of these results is consistent with Table 1, with all tests showing higher entropy in the control trajectories than those of the schizophrenia patients. The consistency across multiple tests suggests the high robustness of the proposed methodology.

	Regression		Student’s *t*-Test		Kolmogorov–Smirnov Test		Permutation Test	
	*t*-Statistic	*p*-Value	*t*-Statistic	*p*-Value	*KS*-Statistic	*p*-Value	Hodges’ *G*	*p*-Value
tFNCP 1	1.00243	0.31694	NaN	N/A	0.12219	0.18315	0.16623	0.14119
tFNCP 2	0.11107	0.91164	−0.04757	0.96209	0.04706	0.99435	−0.00539	0.96050
tFNCP 3	0.78467	0.43327	0.99200	0.32197	0.09868	0.41693	0.11238	0.32237
tFNCP 4	−3.59408	0.00038	−3.83671	0.00015	0.20219	0.00288	−0.43413	0.00030
tFNCP 5	−5.37432	1.55 × 10^−07^	−5.31133	2.09 × 10^−07^	0.27111	1.55 × 10^−05^	−0.60149	1.00 × 10^−04^
tFNCP 6	−4.67690	4.41 × 10^−06^	−4.57202	7.00 × 10^−06^	0.28241	5.71 × 10^−06^	−0.51771	1.00 × 10^−04^
tFNCP 7	−5.01616	9.04 × 10^−07^	−5.12340	5.30 × 10^−07^	0.31515	2.49 × 10^−07^	−0.58108	1.00 × 10^−04^
tFNCP 8	−5.68322	3.13 × 10^−08^	−5.78981	1.73 × 10^−08^	0.29528	1.74 × 10^−06^	−0.65556	1.00 × 10^−04^
Joint Entropy	−5.49083	8.53 × 10^−08^	−5.48028	8.82 × 10^−08^	0.31544	2.42 × 10^−07^	−0.62058	1.00 × 10^−04^

## Data Availability

The network templates used in this paper are available online at www.yuhuidu.com and http://trendscenter.org/software, accessed on 5 December 2023. The code used in this article is available at https://github.com/davidblair8/Entropy-intro, accessed on 1 May 2024.

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
