# Peer review of "A Dynamic Entropy Approach Reveals Reduced Functional Network Connectivity Trajectory Complexity in Schizophrenia"

_entropy, 2024, doi:10.3390/e26070545_

Round 1

Reviewer 1 Report

Comments and Suggestions for Authors

The authors presented a new method to quantify entropy in functional connectivity-based analyses. The method was applied to an fMRI dataset composed of controls and patients with schizophrenia. The good results of their analyses support the usefulness of the proposed method.

The manuscript is very well written, which makes it very enjoyable to read. The methods seem solid and well-described. This, along with the code repository they provide, greatly facilitates the possibility of reproducibility.

For all of the above, my recommendation is the acceptance of the article, subject to the improvement of the minor aspects that I indicate below:

- In the introduction, I miss references to articles that evaluate Shannon entropy in EEG recordings in schizophrenia during a cognitive task (that is, dynamically, in different time windows). Although this study is fMRI, the authors may want to consider mentioning them. The studies by Dr. Vicente Molina's research group are an example of them, although I believe they are not the only ones. I leave it open for the authors to consider whether any of these articles are useful to describe the background of the study.

- Although it is not essential, I think it would be beneficial to explicitly state the objective of the study in the introduction.

- The figures (and this I think would be my main criticism) could be improved:

Fig.1: The font size is not appropriate in many cases. The processing workflow is not followed well. Although I understand that the authors have wanted to detail each step in great detail, the titles of each step are not very representative and, in many cases, are not well understood because at this point they have not yet been explained. I recommend integrating more schematic and easy-to-understand information.

Fig.2: Excessively large figure with font that is too large (title) or too small (rows/columns). It is not indicated what the colors represent in the colorbar (although it is in the caption).

Fig.3: In panel E, would it be appropriate to indicate the p-values ​​instead of the difference of values? An R difference of 0.2 may or may not be significant...

Table.1: The authors may want to consider using a bar figure and indicating significant differences. I don't quite understand why beta was used instead of statistical significance values ​​(p-values).

Fig.4: Although columns A and B are visually very attractive, the size makes them completely useless for the reader. Column C summary would be the most useful, but I can't even distinguish well which boxplot corresponds to controls and which to patients.

- Finally, I do not know if the authors have the medication doses for each patient. If available, it should be treated as a covariate in all analyses. If they do not have them, I consider it a limitation of the study (unless the patients have had a period prior to registration without taking medication).

Author Response

In the introduction, I miss references to articles that evaluate Shannon entropy in EEG recordings in schizophrenia during a cognitive task (that is, dynamically, in different time windows). Although this study is fMRI, the authors may want to consider mentioning them. The studies by Dr. Vicente Molina's research group are an example of them, although I believe they are not the only ones. I leave it open for the authors to consider whether any of these articles are useful to describe the background of the study.

  • I have added two references from Dr. Molina to the introduction (lines 69-70)

Although it is not essential, I think it would be beneficial to explicitly state the objective of the study in the introduction.

  • The first two sentences of the last paragraph in the introduction (lines 83-87) have been rewritten in order to emphasize the present article’s goals.

Finally, I do not know if the authors have the medication doses for each patient. If available, it should be treated as a covariate in all analyses. If they do not have them, I consider it a limitation of the study (unless the patients have had a period prior to registration without taking medication)

  • Regrettably, the FBIRN dataset used in this study does not include medication dosages.

Fig.1:

  • The font size is not appropriate in many cases.
    • I am not certain which sections of the figure are inappropriately sized. Could the reviewer provide further explanation?
      • Which sections of the figure are of inappropriate size?
      • Are these sections too large or too small?
    • The processing workflow is not followed well. Although I understand that the authors have wanted to detail each step in great detail, the titles of each step are not very representative and, in many cases, are not well understood because at this point they have not yet been explained. I recommend integrating more schematic and easy-to-understand information.
      • Figure 1 has been moved to the end of the methods section, following the detailed explanation of each step in the process.
      • Figure 1 has been visually divided and labeled to follow the section titles of the methods section.

Fig 2:

  • Excessively large figure with font that is too large (title) or too small (rows/columns).
    • Text sizes have been adjusted on both title and row/column labels
  • It is not indicated what the colors represent in the colorbar (although it is in the caption).
    • A legend has been added to the colorbar

Fig 3: In panel E, would it be appropriate to indicate the p-values ​​instead of the difference of values? An R difference of 0.2 may or may not be significant...

  • While p-values would be necessary if this article were searching for group-level static connectivity alterations, this is not the case. Figure 3 serves two purposes:
    • To confirm that functional connectivity is properly reconstructed (panel A)
    • To illustrate the significantly depressed joint entropy of patient dynamic functional connectivity
  • The authors did not intend to offer additional statistical evidence for the dysconnectivity hypothesis in Panels C-D, only to provide a visual “sanity check” that the FBIRN dataset does not contradict this established hypothesis. If the reviewer feels that statistical evidence could strengthen the authors’ position, however, we can attempt to implement such a comparison.

Fig 4: In panel E, would it be appropriate to indicate the p-values ​​instead of the difference of values? An

  • Although columns A and B are visually very attractive, the size makes them completely useless for the reader.
    • Authors have substantially increased size of figure legends and labels in column A
    • Authors removed the connectogram column in order to increase size and visibility of connectivity matrices and boxplots.
  • Column C summary would be the most useful, but I can't even distinguish well which boxplot corresponds to controls and which to patients.
    • Boxplot labels and legends have been substantially increased in size.

Table 1: the authors may want to consider using a bar figure and indicating significant differences. I don't quite understand why beta was used instead of statistical significance values ​​(p-values).

  • Significant differences are reported in Figure 3 (joint entropy) and Figure 4 (tFNCP entropies). b serves to illustrate the magnitude and direction of the alteration.  Since controls were assigned a contrast value of 0 and patients a contrast value of 1, a negative b  indicates depressed entropy in patients.

Table 2:

  •  

Reviewer 2 Report

Comments and Suggestions for Authors

This study is original and interesting. The abstract is too long, and, in general, the text can be reduced. The results should be better presented.

Comments on the Quality of English Language

Good quality, only minor corrections are needed.

Author Response

The abstract is too long, and, in general, the text can be reduced.

  • abstract has been shortened
  • a general review has attempted to reduce word count

The results should be better presented.

  • Main author is uncertain how to implement this advice.  Further advice requested.

Reviewer 3 Report

Comments and Suggestions for Authors

Review for manuscript entropy-3015457-v1 “A dynamic entropy approach reveals reduced functional network connectivity trajectory complexity in schizophrenia” by D.S. Blair.

            This manuscript reports a study of the relationship between brain entropy and schizophrenia quantified using information theoretic measures applied to an estimated of subject’s dynamic functional connectivity state space. The authors develop an novel method to identify a basis for the dynamic functional connectivity state space estimated via fMRI. Over the course of the scan, they then track and assess the dynamic trajectory through this basis space. The authors find that the trajectories of schizophrenia patients are less complex than those of a healthy control group along specific dimensions of the basis space, with a connection between entropy generation and cognitive performance. The authors conclude that their findings indicate the usefulness of applying dynamic systems theory to neuroimaging metrices of psychiatric and neurological disorders, as well as normal neurocognitive states.

            I found this study to present a fascinating and potentially very valuable approach to the study of neurocognition. I have only a few comments/questions about the experimental and analytic methodology (see below), as overall I find it to be very clear and well-reasoned. My minor comments/questions are given below:

p.3, Figure 1: The figure caption could be modified to improve the figure’s understandability. For example, it would be helpful on line 116 to state that the sliding window are of length T. On lines 117 – 119, it would also be helpful to explicity state how the matrix V_lxk in the figure relates to the statement “The upper triangle of each connectivity matrix is vectorized to produce 𝑁𝑠𝑢𝑏𝑗𝑒𝑐𝑡𝑠 𝑙×𝑇 subject-level time-resolved connectivity records”. Finally, in the figure, the acronym tFCNP is used yet it has not yet been defined in the figure caption or the main text of the manuscript.

p.5, lines 194 – 195: The estimates of goodness of fit for the different model orders should be reported. Also, how was goodness of fit calculated explicitly? The authors mention a couple of possibilities but do not say which method they used specifically.

p.6, line 230: For clarity, the formula for the KL estimator should be explicitly stated here.

p.7, Figure 2: For clarity, a colorbar indicating the data scale should be included in the figure.

p.7 – 8, Section 3.1: How was the entropy of the sFNCs computed?

p.11, Table 2: This table is a bit confusing and can be reorganized. For example, the directionality of the regression coefficient and its t-value depend on how one organizes the data from the different groups. How this is done is unclear from this table and the associated text in the figure caption and manuscript. Also, are the joint entropy values given in this table collapsed across the two subject groups? Moreover, the authors should state why these rates are collapsed across the last five tFCNPs, i.e., they are the one’s that are statistically significant.

p.13, lines 447 – 449: The authors state “First, no tFNCP displays strong intra-domain correlation. High correlation values instead concentrate between domains, as seen in the cerebellar-subcortical block of tFNCP three (3)”. Is this really that surprising given that the authors used ICA to create the functional basis space? ICA tends to favor interactions between sources rather than within sources. The ICA method used here maps subject data to preestablished network templates that span a variety of brain regions, thus the “sources” identified here are intrinsically multi-domain, correct? Could this the be a limitation of the author’s approach?

Author Response

p.5, lines 194 – 195: The estimates of goodness of fit for the different model orders should be reported. Also, how was goodness of fit calculated explicitly? The authors mention a couple of possibilities but do not say which method they used specifically.

  • Additional explanation has been added to this section (lines 168-170)

line 230: For clarity, the formula for the KL estimator should be explicitly stated here.

  • Equations have been added to this section (lines 204-224)

Section 3.1: How was the entropy of the sFNCs computed?

  • The joint entropy refers to the joint entropy of the group-level dynamic functional connectivity, not the static functional connectivity. Section 3.2 and the caption of Figure 3 has been modified to clarify this.

The authors state “First, no tFNCP displays strong intra-domain correlation. High correlation values instead concentrate between domains, as seen in the cerebellar-subcortical block of tFNCP three (3)”.

  • Is this really that surprising given that the authors used ICA to create the functional basis space?
    • Each individual NeuroMark functional network (FN) is a separate source. Each FN was assigned to one of seven functional domains via comparison to anatomical and functional studies.  For instance, according to the NeuroMark system, the default-mode network (DM) is a domain which consists of seven distinct sources (FNs).  Prior NeuroMark literature, as well as the static FNC reported in Figure 2, usually reports high correlation between FNs in the same functional domain.  These prior reports do not employ temporal ICA on the NeuroMark spatial template, however.
  • ICA tends to favor interactions between sources rather than within sources.
    • Perhaps I am misunderstanding something, but I believe that independent component analysis is supposed to minimize dependencies between sources. Two signals which predictably interact should create a dependency, and ICA should seek to minimize that dependency by combining the signals into a single source, no?
    • Main author is requesting advice from last author on this point.
  • The ICA method used here maps subject data to preestablished network templates that span a variety of brain regions, thus the “sources” identified here are intrinsically multi-domain, correct? Could this the be a limitation of the author’s approach?
    • It is true that each tFNCP consists of a linear combination of NeuroMark sources and can thus be expected to contain signals from multiple domains. It is somewhat surprising, however, that functionally related sources (sources from the same functional domain) appear to interact with each other less predictably than sources from different functional domains.  This is all the odder given that static FNC consistently shows higher intra-domain correlation than inter-domain correlation (Figure 3).

Fig.1:

  • The figure caption could be modified to improve the figure’s understandability. For example, it would be helpful on line 116 to state that the sliding window are of length T.
    • The window length and number of windows has been clarified (length = 40, count = T).
  • On lines 117 – 119, it would also be helpful to explicity state how the matrix V_lxk in the figure relates to the statement “The upper triangle of each connectivity matrix is vectorized to produce ??????????×? subject-level time-resolved connectivity records”.
    • An additional legend has been added to this effect
  • Finally, in the figure, the acronym tFCNP is used yet it has not yet been defined in the figure caption or the main text of the manuscript.
    • Figure 1 has been moved to the end of the methods section; as such, tFNCP has been defined before it is encountered.

Fig 2: For clarity, a colorbar indicating the data scale should be included in the figure.

  • A colorbar and legend have been added to this figure

Table 2: This table is a bit confusing and can be reorganized.

  • For example, the directionality of the regression coefficient and its t-value depend on how one organizes the data from the different groups. How this is done is unclear from this table and the associated text in the figure caption and manuscript.
    • Regression coefficients are reports in Table 1, not Table 2.
    • Controls were assigned a contrast label of 0, patients a contrast label of 1. A negative regression coefficient thus indicates depressed entropy in patients.  Both table captions have been updated to clarify this.
  • Also, are the joint entropy values given in this table collapsed across the two subject groups? Moreover, the authors should state why these rates are collapsed across the last five tFCNPs, i.e., they are the one’s that are statistically significant.
    • There appears to be some confusion. Table 2 does not report entropy values; it reports the results of statistical tests on entropy distributions (p-values and test statistics).  As such, there is no “collapse” of entropy values.

Round 2

Reviewer 3 Report

Comments and Suggestions for Authors

The changes the authors have made are satisfactory. I apologize to the authors for my confusion on certain points of the manuscript and thank them for their clarification. Regarding my statement that “ICA tends to favor interactions between sources rather than within sources”, this is a misstatement on my part; the authors are correct that ICA minimizes dependencies between sources. My initial concern here was based on a confusion of the author’s usage of the terms “source” and domain. With the authors clarification of these terms in their reply, my concern no longer applies.